# Overestimated cytotoxicity and underestimated whitening efficacy of glabridin: A result of its poor solubility in DMSO

Haiyan Liu[1], Anning Wang[1], Xiaoyi Chen[2], Sen Hou ![ORCID][1,3]*, Anzhang Li[1,3]*

1 Guangzhou Fanzhirong Cosmetics Co., Ltd., Huangpu District, Guangzhou, Guangdong, China, 2 Guangzhou Zhongkejian Technology Testing Co., Ltd., Guangzhou Fanzhirong Cosmetics Co. Ltd., Huangpu District, Guangzhou, Guangdong, China, 3 Guangzhou Qingnang Biotechnology Co., Ltd., Guangzhou Fanzhirong Cosmetics Co. Ltd., Huangpu District, Guangzhou, Guangdong, China

* housen@guyu.com.cn (SH); lianzhang@guyu.com.cn (AZL)

## Abstract

Glabridin is widely used as a whitening agent in cosmetics, but its cytotoxicity remains a key concern in safety evaluations. In typical cytotoxicity assays, glabridin is dissolved in dimethyl sulfoxide (DMSO) before being added to the cell culture medium because it is insoluble in water. However, our study revealed that when the DMSO solution of glabridin was mixed with cell culture medium, glabridin was rapidly released due to its poor solubility in the DMSO/water mixture. The released glabridin rapidly formed crystals, which failed to enter cells. Consequently, the whitening efficacy of glabridin was reduced. Moreover, the glabridin crystals produced higher cytotoxicity, possibly due to the physical damage caused by their sharp crystalline structures. However, encapsulating glabridin in cyclodextrin (CD) can address these challenges, offering a better approach for glabridin cytotoxicity assays. The CD encapsulation method, compared to the DMSO solution method, not only decreased the cytotoxicity of glabridin but also increased its whitening efficacy. By comparing the efficacy of glabridin dissolved in DMSO and encapsulated in CD, we discovered that the reported cytotoxicity of glabridin may have been overestimated in previous cytotoxicity studies which used DMSO as a solvent, while its whitening efficacy may have been underestimated. These findings not only offer new insights for in vitro studies of glabridin-like reagents, but also facilitate the development of safer and more effective whitening products.

## Introduction

Glabridin, an important cosmetic ingredient, was first discovered and isolated from the roots of *Glycyrrhiza glabra* L. in 1976 [1]. Due to its excellent anti-inflammatory, antioxidant, antibacterial, and whitening efficacy [1–4], glabridin is now widely utilized in cosmetic formulations. Importantly, compared to other whitening ingredients

**Data availability statement:** All relevant data are within the manuscript and its Supporting Information files.

**Funding:** This study was supported by a scientific grant from Guangzhou Fanzhirong Cosmetics Co., Ltd. (MT-2023-002).The funder had no role in study design, data collection and analysis, decision to publish, or preparation of the manuscript.

**Competing interests:** The authors have declared that no competing interests exist.

such as arbutin [5] and nicotinamide [6], glabridin shows outstanding efficiency in suppressing melanin synthesis. Glabridin significantly inhibits the expression of tyrosinase (TYR, TYR-1, TYR-2) in B16 cells and alleviates UVB-induced skin pigmentation in guinea pigs [7,8]. Furthermore, glabridin regulates melanin production by inhibiting pathways such as the cAMP/PKA/CREB/MAPK/MITF signaling pathway [9]. However, despite these benefits, safety concerns have emerged due to reports regarding the strong cytotoxicity of glabridin. Glabridin shows poor solubility in both oil and water [10] and this cannot be added directly to the culture medium during mammalian cell assays. The solubility characteristics of glabridin may potentially affect both its cytotoxicity and whitening efficacy.

Dimethyl sulfoxide (DMSO), a highly polar solvent, is commonly used in cell and tissue experiments due to its good solubility for most water-insoluble chemicals [11]. Owing to the poor solubility, glabridin is often dissolved in DMSO in cytotoxicity assays [9,12]. Typically, glabridin is first dissolved in DMSO, and the solution is then used to treat the target cells. However, when the DMSO-glabridin mixture is diluted with cell culture medium, the concentration of DMSO decreases, leading to the precipitation of previously dissolved glabridin [13].In addition, these precipitants can form crystals, which may induce cellular damage. Several studies show that the physical characteristics of these crystals, such as their size and shape, influence their cell toxicity and uptake behavior [14,15]. Sun et al. demonstrated that sharp crystals can easily cause cell membrane rupture, while blunt crystals may reduce cellular damage [16].

Studies on the cytotoxicity of glabridin have yielded varying outcomes. For example, Cao et al. showed that glabridin at a concentration of 8.76 µg/mL led to an apoptosis rate of 32.25% in B16-F10 cells, with an IC50 of approximately 16.82 µg/mL [12]. Meanwhile Pan et al. reported that glabridin of concentrations below 10.51 µg/mL had no significant effect on B16 cell viability [9]. Wang et al. reported that glabridin had minimal effects on the activity of B16 melanocytes within the concentration range of 3.125–50 µg/mL [17]. Similarly, reports regarding the whitening effect of glabridin also vary considerably across different studies [7–9,17]. These differences in the reported cytotoxicity and efficacy of glabridin could be attributed to the precipitation of this compound from DMSO solutions and the consequent formation of crystals that cannot cross the cell membrane and thus provide weak whitening effects.

Cyclodextrin (CD), an oligosaccharide composed of 6–8 D-glucopyranose units, has several benefits including low toxicity and ease of modification. By forming a super-molecular complex with glabridin, CD can significantly enhance its solubility and stability [18]. Moreover, since glabridin is encapsulated by CD through intermolecular interaction, its stability is not influenced by dilution.

In this study, we investigated the effect of dilution-induced glabridin precipitation and crystallization in DMSO on its cytotoxicity and whitening efficacy. Specifically, the solubility, stability, cell toxicity, and melanin synthesis inhibition of glabridin dissolved in DMSO was compared with that of glabridin encapsulated in CD after dilution. The implication of our findings was also discussed with regard to the overestimation of glabridin cytotoxicity and the underestimation of its whitening efficacy.



## Materials and methods

### Materials

Glabridin was purchased from Guangzhou Qingnang Biotechnology Co., Ltd. DMSO (analytical grade) was purchased from Merck KGaA. 2-hydroxylpropyl-β-cyclodextrin (HP-β-CD) was purchased from Shandong Binzhou Zhiyuan Biotechnology Co., Ltd. Phosphate-buffered saline (PBS) buffer solution (pH 7.4) and RPMI 1640 were purchased from Thermo Fisher Biochemistry (Beijing) Co., LTD. Ethanol (analytical grade) was purchased from Merck KGaA. Methanol (chromatographic grade) was purchased from Shanghai Aladdin Biochemical Technology Co., Ltd. Dulbecco's Modified Eagle's Medium (DMEM; high glucose) and 3-(4,5-dimethylthiazol-2-yl)-2,5-diphenyltetrazolium bromide (MTT) were purchased from Shanghai XP Biomed Ltd. Fetal bovine serum (FBS) and trypsin were purchased from Gibco Life Technologies.

### Preparation of glabridin solutions (DMSO and CD)

A solution of glabridin in DMSO was prepared by mixing glabridin powder with DMSO and stirring for 10 min at room temperature. Meanwhile, to prepare CD-encapsulated glabridin, a mixture of CD, glabridin and water (1:9:10) was stirred at 60 °C for 12 h. The mixture was then incubated at 60 °C in a vacuum incubator (−0.06 mPa) for 8 h to obtain CD-encapsulated glabridin powder. The powder was dissolved in PBS to obtain glabridin solutions.

The drug loading efficiency (DLE) was determined by Eq 1:

$$DLE = \frac{C_1}{C_0} \times 100\%$$

(1)

where $C_0$ represents the total mass of CD-glabridin complex, and $C_1$ represents the mass of glabridin loaded within the complex, as determined by HPLC analysis of glabridin content.

### Stability assay

Glabridin was dissolved in DMSO to prepare a 20 mg/mL solution. Similarly, a 20 mg/mL solution of CD-encapsulated glabridin was also prepared in PBS. Both solutions were diluted with PBS to a final concentration of 100 µg/mL. These dilutions were incubated at room temperature for 0, 5, 10, 15, 30, 60, and 120 min, respectively. The solutions were filtered using a 0.22-µm filter and then diluted with 9 times the volume of methanol. Then, the concentration of glabridin in the supernatant was quantified using a ZORBAX Stable Bond C18 column (4.6 × 250 mm, 5 µm, Agilent Technologies) linked to a high-performance liquid chromatography (HPLC) system equipped with a diode array detector (DAD, Agilent 1100). The column temperature was maintained at 25 °C, and the flow rate was maintained at 1 mL/min. The mobile phase consisted of acetonitrile and water (4:1 v:v). Glabridin with a purity of 99.7% (FUJIFILM Wako Pure Chemical Corporation, Japan) was used as the standard [19].

For another set of experiments, glabridin was dissolved in DMSO to prepare a 20 mg/mL solution. Meanwhile, a 20 mg/mL solution of CD-encapsulated glabridin was also prepared in PBS. Both solutions were serially diluted to concentrations of 6.25, 12.5, 25, 50 and 100 µg/mL, respectively using PBS. These solutions were filtered with a 0.22-µm filter and then diluted with 9 times the volume of methanol. Finally, the concentration of glabridin in the supernatants was quantified as described above.

The content retention ratio (R), which represented the fraction of soluble glabridin after dilution, was calculated using Eq. 2 as follows:

$$R = \frac{C_1}{C_0} \times 100\%$$

(2)

where $C_0$ represents the concentration of glabridin before dilution, and $C_1$ represents the concentration of glabridin after dilution.



## Dynamic observation of glabridin precipitation

Glabridin was dissolved in DMSO to prepare a 20 mg/mL solution, which was diluted to a final concentration of 100 μg/mL with PBS. This solution was placed both in a beaker and in the well of a microscopy slide with recess. The mixtures were incubated at room temperature for 0, 10, 30, 60, and 120 min, respectively. The solution in the beaker was imaged using a DSLR camera. Meanwhile, the solution on the slide was observed under a polarizing microscope (Labomed, USA) equipped with the TCapture software.

## Cell viability assay

B16-F10 cells were routinely cultured in RPMI 1640 medium supplemented with 10% FBS at 37 °C under 5% $CO_2$. Before experiments, the cells were seeded in the wells of a 96-well plate and cultured for 12 h. Glabridin was dissolved in DMSO to prepare a 20 mg/mL solution. The glabridin solution was diluted with cell culture medium containing 10% FBS to a series of concentrations of 0, 0.78, 1.56, 6.25, 12.5, 25, 50 and 100 μg/mL, respectively. The cells were cultured with the glabridin solutions for 24 h and the cell viability was tested using an MTT Kit (VivaCell). The optical density (OD) values were measured at a wavelength of 490 nm using a microplate reader (Tecan, Spark). Untreated cells were used as a control. Blank wells were set as blank samples. The cell viability (CV) was calculated using Eq. 3 as follows:

$$CV = \left(1 - \frac{OD_{sample} - OD_{blank}}{OD_{control} - OD_{blank}}\right) \times 100\%$$

(3)

where $OD_{sample}$ represents the OD value of the tested samples, $OD_{blank}$ represents the OD value of the blank sample, and $OD_{control}$ represents the OD value of the control. The half-maximal inhibitory concentration (IC50) value was calculated using Origin software (2022 version).

A 20 mg/mL solution of CD-encapsulated glabridin in PBS was also prepared. The glabridin solution was diluted with cell culture medium containing 10% FBS to obtain a series of concentrations of 0, 0.78, 1.56, 6.25, 12.5, 25, 50 and 100 μg/mL, respectively. The cell viability after treatment with CD-encapsulated glabridin was measured using the same method described above.

Pure CD and DMSO was used as a control to test the cytotoxicity of the carrier. The concentrations of CD were 144.96, 289.92, 579.84, 1159.69, 2319.38, 4638.75, and 9277.50 μg/mL, respectively. The concentrations of DMSO were 1000, 5000, 10000, 25000, 50000, 100000, 150000, and 200000 μg/mL, respectively.

## Melanin synthesis inhibition assay

Both glabridin dissolved in DMEM and CD-encapsulated glabridin were diluted with cell culture medium containing 10% FBS to concentrations of 0.78 and 3.125 μg/mL. The cells were cultured for 24 h in RPMI 1640 Medium and then the medium was replaced with DMEM containing glabridin at different concentrations. After cell culture with these glabridin solutions for 24 h, the culture medium was removed and the cells were lysed using 1 mol/L NaOH containing 10% DMSO. The mixtures were incubated at 80 °C for 1 h and then transferred to 96-well plates. Cells cultured in RPMI 1640 medium were used as blank control (BC). Cells treated with DMEM without glabridin was used as negative control (NC). The OD values were measured at the wavelength of 405 nm using a microplate reader (Tecan, Spark) [20]. Untreated cells were used as a control. The inhibition of melanin (IM) was calculated using Eq. 4 as follows:

$$IM = \left(1 - \frac{OD_{sample} - OD_{blank}}{OD_{control} - OD_{blank}}\right) \times 100\%$$

(4)

where $OD_{sample}$ represents the OD value of the tested samples, $OD_{blank}$ represents the OD value of the blank well, and $OD_{control}$ represents the OD value of the control.

Pure CD was used as a control to test the whitening effect of the carrier. The concentrations of CD were 7.24 and 28.99 µg/mL, respectively. The concentrations of CD were equivalent to the concentrations of CD in CD-encapsulated glabridin solutions, respectively.

### Statistical analysis

All data were analyzed with SPSS for Windows, version 21.0. Significant differences were identified using the Student's t-test. The significance thresholds were as follows: NS $p > 0.05$, *$p < 0.05$, **$p < 0.01$, and ***$p < 0.001$.

## Results

### Glabridin solubility

Dissolving glabridin in DMSO and encapsulating glabridin within CD can both effectively increase its solubility in water [10,21]. Our findings showed that the solubility limit of glabridin in DMSO is higher than 20 mg/mL. Meanwhile, the results also indicated that glabridin can efficiently enter the hydrophobic cavity of CD, getting trapped within the CD molecule (**Fig 1**). Previous studies have demonstrated that glabridin forms a 1:1 stoichiometric inclusion complex with CD through characteristic host-guest interactions, where the hydrophobic C-ring of glabridin enters the cyclodextrin cavity [10]. This interaction is driven by hydrophobic interactions between the nonpolar cavity and glabridin's lipophilic structure. Meanwhile, the hydrophilic exterior of CD enhances aqueous solubility. The drug loading efficiency of glabridin in CD was calculated to be 9.73%. The specifics of the encapsulation method agrees with Wei's work [10]. The maximum glabridin concentration typically used in cytotoxicity assays is less than 100 µg/mL [9,12,17]. Hence, in this study, we diluted the glabridin solution to a concentration of 100 µg/mL or even lower with PBS to study the precipitation phenomena.

Various formulation strategies have also been applied to improve the solubility and physical stability of poorly soluble ingredients, such as glabridin. For instance, both liposome-based delivery systems and nanoemulsions have been reported to effectively prevent the precipitation of glabridin [22,23]. These systems not only enhance the dispersion of glabridin in aqueous environment but also improve its stability. However, these approaches still suffer from certain



**Fig 1. Schematic diagram showing the process of glabridin encapsulated with CD.**

shortcomings. Nanoemulsions typically use surfactants, which would introduce potential skin irritation [24]. Liposome strategy requires complicated equipment, such as high-pressure homogenizer, and likely introduces additional cytotoxicity. CD encapsulation provides a simpler and more robust approach with favorable formulation compatibility.

## Glabridin precipitation after dilution

Glabridin dissolved in DMSO typically precipitates immediately after dilution. Therefore, we observed that the concentration of glabridin in the DMSO solution decreased significantly after dilution (**Fig 2a**). Even after dilution at a 1:1 volume ratio (**S1 File** in supporting information), the solution started to become turbid, indicating the precipitation of glabridin from the solution. The immediate precipitation of glabridin was linked to the dilution ratio. At a higher dilution ratio, low amounts of glabridin were retained in the solution (**Fig 2b**). This dilution-induced precipitation of glabridin could be attributed to two factors. On one hand, dilution decreased the concentration of DMSO and increased the proportion of water in the solvent mixture. Since glabridin is insoluble in water but soluble in DMSO, this dilution decreased the concentration of soluble glabridin in the DMSO/water mixture. On the other hand, dilution also reduced the total amount of glabridin per unit volume of the DMSO/water mixture. The precipitation of glabridin was a result of this competition: the decrease in the solubility

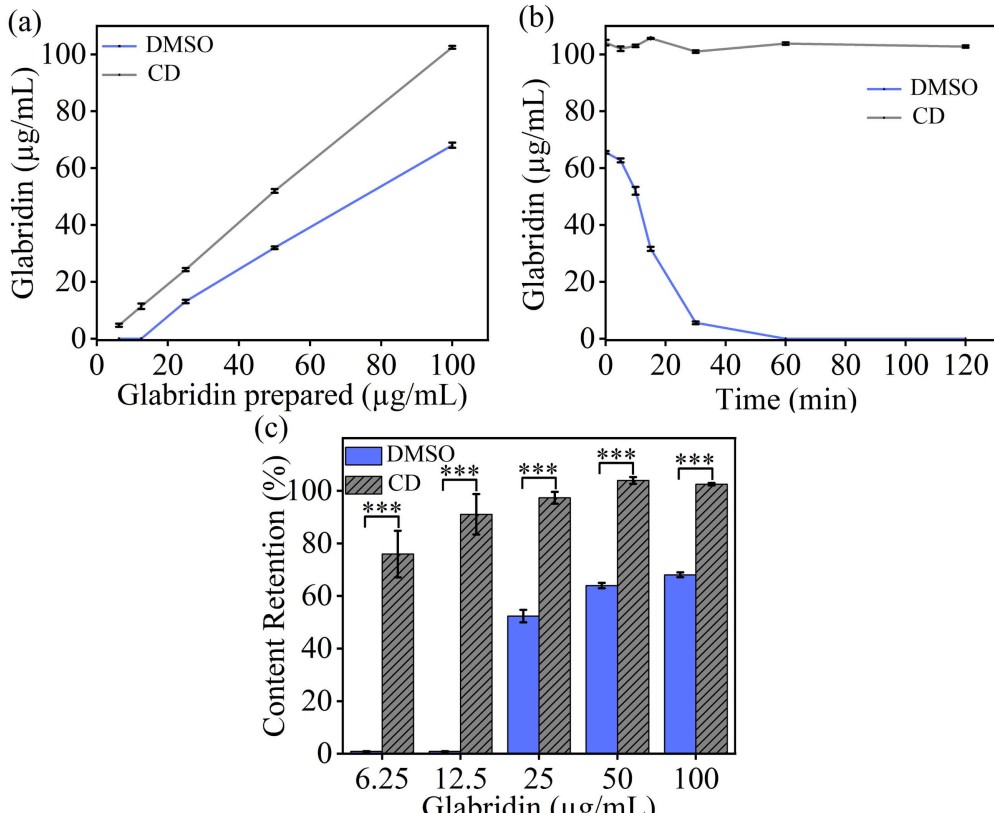

**Fig 2. Changes in glabridin concentrations after dilution.** (a) Glabridin was initially dissolved in DMSO or encapsulated with CD to obtain 20 mg/mL solutions. Then, the solutions were diluted to obtain a series of glabridin concentrations. The concentrations of the remained glabridin were measured immediately after dilution. (b) and (c) Glabridin was initially dissolved in DMSO or encapsulated with CD to obtain 20 mg/mL solutions. Then, the solutions were diluted to 100 μg/mL. The concentrations of the remained glabridin were measured at different elongation time point after dilution. (b) shows the concentration and (c) shows the ratio of the remained glabridin. All data are presented as average value ± SD of triplicates (* $P < 0.05$, **$P < 0.01$, *** $P < 0.001$, **** $P < 0.0001$).

of glabridin outweighed the reduction in its concentration, leading to precipitation. However, no obvious precipitation of CD-encapsulated glabridin was observed during the dilution process. Given that dilution is inevitable during cytotoxicity assays, these results indicated that CD encapsulation is a more effective method for dissolving glabridin than using DMSO as a medium.

We also examined the precipitation process of glabridin over time (**Fig 2b**). The concentration of soluble glabridin in the DMSO solution rapidly decreased from 100 µg/mL to 66 µg/mL. After 60 min, nearly all the glabridin had precipitated out of solution. In contrast, the concentration of CD-encapsulated glabridin remained largely unchanged even after 120 min of dilution. Typically, in the cytotoxicity assays, a series of glabridin solutions are prepared by mixing pure glabridin solutions (e.g., DMSO solutions) with cell culture medium or buffer solutions, and these glabridin solutions are then used to treat cells. The time span between mixing and exposure typically ranges from 10 to 30 min, depending on experimental protocols. Our data indicates that this duration is sufficient to allow the precipitation of glabridin from DMSO solutions (**Fig 2**). Unfortunately, though DMSO is commonly used as a solvent for glabridin in cytotoxicity assays [7,9,17], the potential issues arising from its precipitation have not been considered seriously so far.

In order to observe the precipitation process, the changes in the glabridin/DMSO solution after dilution were studied using polarized light microscopy and imaged with a camera (**Fig 3**). The solution became turbid immediately after dilution. During 2 h of incubation, the precipitates gradually settled at the bottom of the beaker. The turbidity of the glabridin solutions was attributed to the formation of glabridin crystals, which appeared immediately after dilution and were shaped like sharp needles. Both the size and number of these crystals increased with time after dilution.

The formation of glabridin crystals decreased the bioavailability of glabridin. We speculated that this shift would have two major consequences. First, the efficacy of glabridin would be hampered since the precipitated glabridin crystals would fail to effectively enter cells. Additionally, the sharpness of these crystals would induce physical cellular damage, thus increasing the observed cytotoxicity of glabridin. Previous studies showed that the shape and aggregation state of calcium oxalate monohydrate (COM) crystals were crucial factors in their toxicity to renal epithelial cells [16]. Crystals with sharp edges caused more severe cell damage, including membrane rupture, increased ROS production, and mitochondrial dysfunction, leading to necrosis. Aggregation of sharp-edged crystals exacerbated cell injury, while aggregation of blunt-edged crystals showed reduced toxicity. These findings pointed out the significant role of crystal shape and aggregation in cytotoxicity.

## Glabridin cytotoxicity after dilution

We evaluated the cytotoxicity of glabridin by initially dissolving it in DMSO and then diluting this solution with cell culture medium to obtain the required final concentration. During this assay, the glabridin samples were prepared quickly and added to cells immediately without delay. The calculated IC50 of DMSO-dissolved glabridin was 20.56 ± 2.55 µg/mL (**Fig 4a**). Meanwhile, glabridin encapsulated in CD showed an IC50 of approximately 32.61 ± 0.88 µg/mL, representing a 58.61% increase in safety in comparison with the glabridin/DMSO samples (**Fig 4b**). Studies have reported that DMSO concentrations of less than 0.6% are non-toxic to cells [25]. Our result well agreed with the literature. Our data showed that DMSO of a concentration below 0.5% did not affect cell viability (**Fig 4e**), and therefore, the cytotoxicity of the glabridin/DMSO mixture could be attributed to glabridin alone. Similar results were observed with the mixture of glabridin encapsulated in CD (**Fig 4d**). Under these experimental conditions, glabridin underwent precipitation in the cell culture medium, as shown in **Fig 3**. Some of the glabridin was taken up by cells, while the remaining glabridin formed sharp crystals. These sharp crystals could induce physical damage to the cells, thus increasing the cytotoxicity of glabridin. Herein, the cytotoxicity of glabridin was enhanced by almost 50% due to the dilution-induced precipitation. We speculate that this increase in cytotoxicity may not be consistent across different experimental conditions because the precipitation process is time-sensitive, and the time between dilution and cell treatment is difficult to precisely control.

To verify the influence of glabridin precipitation on cytotoxicity, we deliberately prolonged the precipitation time to 0.5 h (**Fig 4c**). Cytotoxicity assay showed that under these conditions, the IC50 value of glabridin was approximately

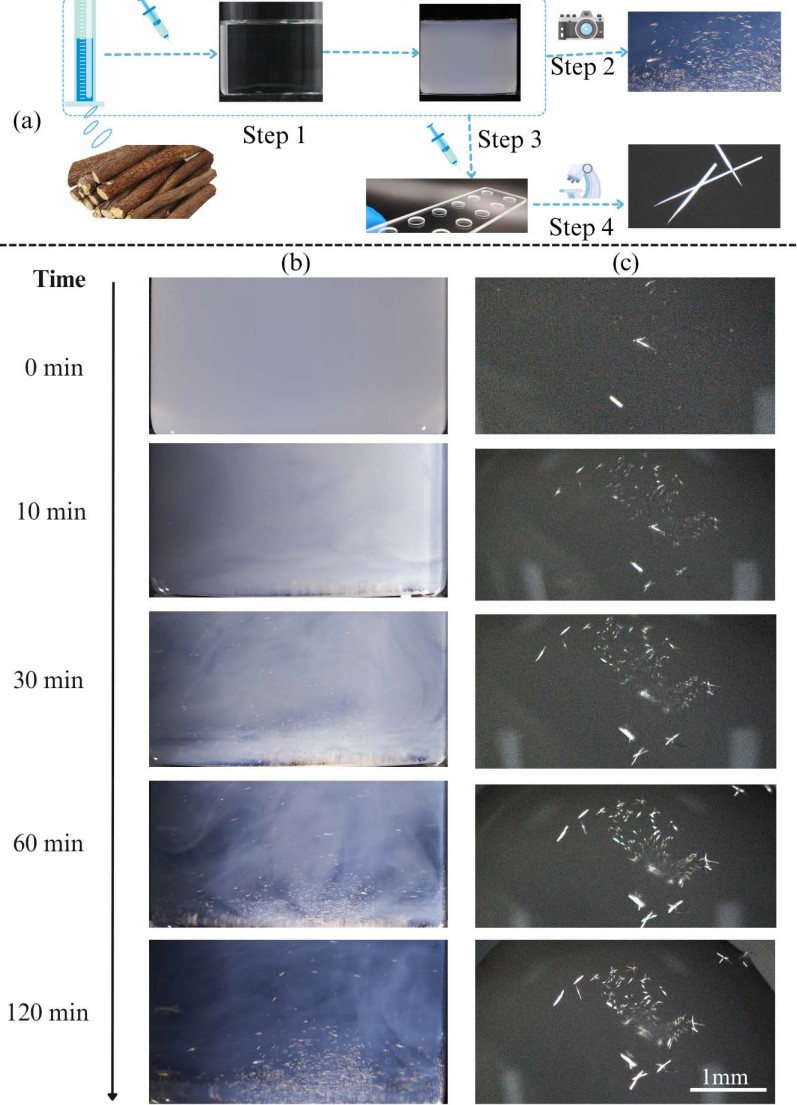

**Fig 3. Precipitation of glabridin from DMSO solutions after dilution with PBS.** (a) Schematic image describing the experimental procedure. (b) Turbidity shift due to glabridin precipitation from DMSO/PBS mixtures at different time points after dilution. (c) Glabridin crystal formation at different time points after dilution.

$3.96 \times 10^{10}$ µg/mL. This indicated that the glabridin exhibited negligible toxicity to cells, and its safety increased by around 1 billion times. We speculated that all the glabridin had precipitated and formed crystals, leading to its bioavailability close to 0. Although the sharp crystal structure of glabridin was expected to cause some physical cellular damage, this damage seemed negligible under the condition that most glabridin cannot enter cells because of precipitation.

Using DMSO as a solvent for glabridin may introduce data vibration in the cytotoxicity assay. Usually, the glabridin solution is added to the cells immediately after preparation. Under these circumstances, the formation of glabridin nano-crystals can induce higher cytotoxicity, leading to an overestimation of cytotoxicity. However, in some cases, the prepared glabridin solutions are incubated for several tens of minutes before being used for cell treatment. In these cases, glabridin can show decreased cytotoxicity. By contrast, encapsulating glabridin within CD can overcome the precipitation

**Fig 4. Time dependent cytotoxicity of glabridin in different solutions.** (a, c) shows the cytotoxicity of DMSO dissolved glabridin and (b) shows the cytotoxicity of CD-encapsulated glabridin. The time span between sample preparation and cell exposure was 0 min (a) and 30 min (c), respectively. (d) shows the cytotoxicity of CD, (e) shows the cytotoxicity of DMSO. The curves in a-c are fitting curves for corresponding IC50 calculation. All data are presented as average value ± SD of triplicates (* $P < 0.05$, **$P < 0.01$, *** $P < 0.001$, **** $P < 0.0001$).

challenges caused by dilution. Thus, we propose that CD encapsulation is a more reliable method for glabridin sample preparation in cytotoxicity assay.

Although previous studies have well established cyclodextrin's role in enhancing the bioavailability of poorly soluble compounds [10], our work still makes a key advance beyond existing bioavailability researches. In this study, CD was simply used as a control to show that if the solubility problem of DMSO was avoided by using CD instead, the overestimated

cytotoxicity would not exist. Herein, the cytotoxicity of glabridin might be overestimated by previous studies using DMSO as the solvent.

## Inhibition of melanin synthesis by glabridin

We compared the whitening efficacy of glabridin using two formulations: DMSO-dissolved glabridin and CD-encapsulated glabridin. The concentrations of glabridin were adjusted to maintain a cell viability of more than 88%. Control experiments confirmed that both the CD and DMSO alone had a minimal impact on melanin synthesis (**Figs 5a** and **5b**), so the inhibition effect resulted from glabridin itself.

To minimize the impact of reduced bioavailability due to prolonged incubation, we prepared mixtures of DMSO/glabridin and culture medium and used them for cell treatment without any delay. Both DMSO-dissolved glabridin and CD-encapsulated glabridin showed a significant melanin inhibition capacity (**Fig 5**). After UV irradiation, the melanin synthesis increased by approximately 3 folds. Both DMSO-dissolved and CD-encapsulated glabridin could effectively decrease the melanin synthesis in a dose-dependent manner. Melanin synthesis inhibition rates of nearly 28–42% were achieved after the addition of glabridin, and these rates increased with elevations in glabridin concentration. At the same glabridin concentration, CD-encapsulated glabridin showed a higher inhibition efficiency than DMSO-dissolved glabridin. These results indicated that CD encapsulation did not interfere with the functional properties of glabridin. Since the precipitation of glabridin was inevitable, the concentration of glabridin in DMSO solutions decreased considerably, and the crystals produced were too big to effectively enter the cells. As a result, the melanin inhibition efficiency was decreased.

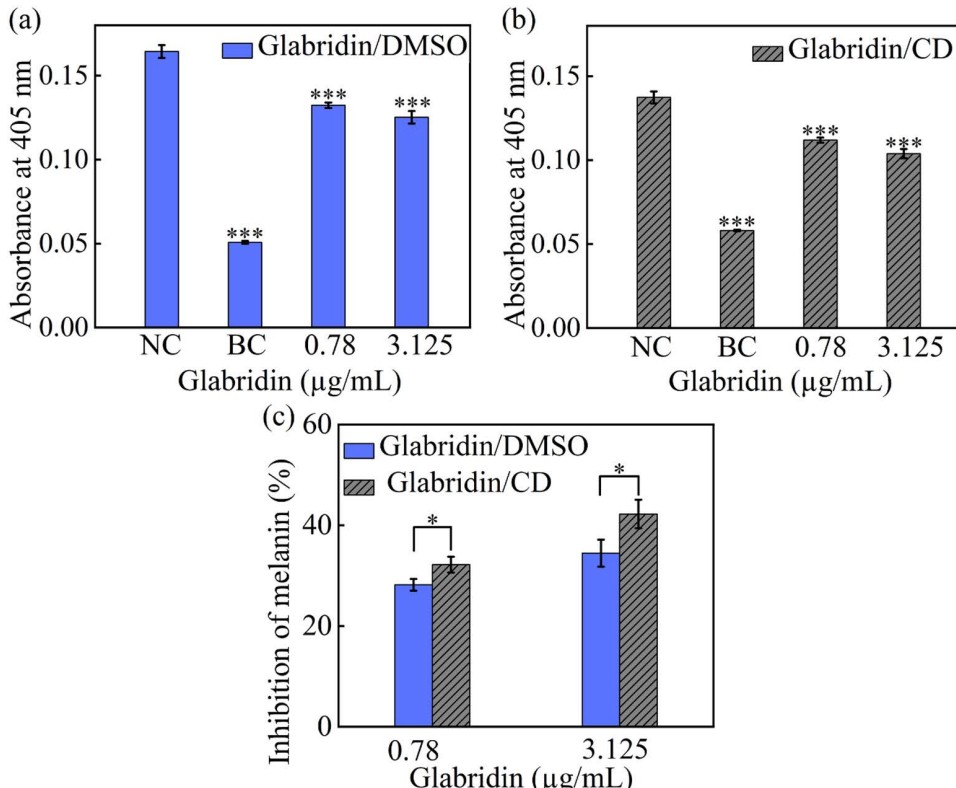

**Fig 5. Melanin synthesis (a, b) and Inhibition of melanin (c) after DMSO-dissolved and CD-encapsulated glabridin treatment (BC: blank control; NC: negative control).** All data are presented as average value±SD of triplicates (* P<0.05, **P<0.01, *** P<0.001, **** P<0.0001).

Because we minimized the incubation time for glabridin precipitation in the present study, the difference in melanin inhibition between DMSO-dissolved and CD-encapsulated glabridin was not very large. However, the precipitation of DMSO-dissolved glabridin still prevented the effective uptake of glabridin. Thus, the experiments using CD-encapsulated glabridin more accurately reflected the whitening efficacy of glabridin and were more meaningful for guiding its potential practical applications.

## Discussion

Glabridin, known for its potential tyrosinase inhibitory activity, is widely used in skincare products owing to its whitening efficacy [26]. Numerous studies have employed DMSO as a solvent to improve the solubility of glabridin in water, facilitating its use in *in vitro* cellular assays. However, though the purity of glabridin in previous studies has been at least 98%, these studies have reported considerable variations in the safe concentrations and efficacy of glabridin [9,12,17]. The present study revealed that glabridin dissolved in DMSO quickly precipitates upon dilution, reducing the actual concentration of soluble glabridin available for cellular uptake. Furthermore, the crystalline form of glabridin in solution can potentially damage cell membranes, skewing the results of cell toxicity and bioactivity assessments [14–16]. Therefore, the choice of solvent can significantly affect the outcome of cell-based experiments. However, the existing literature lacks a systematic analysis of how solvent choice influences the observed cytotoxicity and whitening efficacy of glabridin.

By itself, CD does not exhibit whitening effects and is highly safe [27]. Owing to its unique molecular structure — including its hydrophobic inner cavity and hydrophilic outer shell — CD is commonly used in the cosmetics industry to encapsulate unstable or poorly water-soluble active ingredients [28]. In this study, we compared the solubility, stability, and effects of glabridin in DMSO-dissolve and CD-encapsulated glabridin on cytotoxicity and melanin synthesis in B16-F10 cells. The results showed that DMSO-dissolved glabridin rapidly precipitates upon dilution, forming crystals that increase the risk of cell membrane damage. This leads to reduced cell viability and unstable glabridin concentrations. In contrast, glabridin encapsulated in CD shows significantly improved water solubility and solution stability, preventing potential cellular damage due to crystal precipitation and yielding consistent toxicity results. In this study, CD-encapsulated glabridin exhibited a melanin synthesis inhibition rate 14.51% and 22.58% higher than DMSO-dissolved glabridin at concentrations of 0.78 µg/mL and 3.125 µg/mL, respectively. This suggests that the DMSO dissolving method may overestimate the cytotoxicity of glabridin and underestimate its melanin inhibition efficacy. By contrast, CD encapsulation more accurately reflects the bioactivity of glabridin.

The aim of our study was to highlight the overestimated cytotoxicity and underestimated whitening efficacy of glabridin observed in cell-based experiments. We do not believe that such overestimation and underestimation occur in *in vivo* studies. These effects were caused by precipitation resulting from the dilution process. In *in vitro* experiments, dilution of glabridin is inevitable. However, in *in vivo* conditions, glabridin does not undergo the same dilution process. Therefore, the overestimated cytotoxicity and underestimated whitening efficacy are unlikely to occur in *in vivo* studies.

## Conclusion

In summary, this study provides experimental data for optimizing the use of glabridin in skin whitening applications. Compared to traditional DMSO-dissolved glabridin mixtures, the CD-encapsulated glabridin can reduce the experimental variation caused by experimental conditions, significantly improve the water solubility and stability of glabridin, and more accurately reflect its actual bioactivity. These findings not only offer novel insights for future *in vitro* studies on glabridin based products, but also lay a solid foundation for the development of safer and more effective whitening products. More effort is needed to explore the functional properties of glabridin-like reagents using CD encapsulation as a method for solubility improvement.



## Supporting information

**S1 File. Supporting information and raw data.**
(DOCX)

## Author contributions

**Conceptualization:** Sen Hou.

**Data curation:** Sen Hou.

**Funding acquisition:** Anning Wang.

**Investigation:** Haiyan Liu, Xiaoyi Chen.

**Methodology:** Haiyan Liu.

**Supervision:** Sen Hou, Anzhang Li.

**Writing – original draft:** Haiyan Liu.

**Writing – review & editing:** Sen Hou.

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
