## [Decision Letter · Decision Letter 0]

30 Mar 2025

PONE-D-25-01303Overestimated cytotoxicity and underestimated whitening efficacy of glabridin: A result of its poor solubility in DMSOPLOS ONE

Dear Dr. Sen Hou,

Thank you for submitting your manuscript to PLOS ONE. After careful consideration, we feel that it has merit but does not fully meet PLOS ONE’s publication criteria as it currently stands. Therefore, we invite you to submit a revised version of the manuscript that addresses the points raised during the review process.

We look forward to receiving your revised manuscript.

Kind regards,

Yusuf Ahmed Haggag, PhD

Academic Editor

PLOS ONE

Journal Requirement

2. Thank you for stating the following financial disclosure: This study was supported by a scientific grant from Guangzhou Fanzhirong Cosmetics Co., Ltd (MT-2023-002).  

3. Thank you for stating the following in the Acknowledgments Section of your manuscript: This study was supported by a scientific grant from Guangzhou Fanzhirong Cosmetics Co., Ltd. (MT-2023-002).

This study was supported by a scientific grant from Guangzhou Fanzhirong Cosmetics Co., Ltd (MT-2023-002).

Reviewers' comments:

Reviewer's Responses to Questions

**Comments to the Author**

1. Is the manuscript technically sound, and do the data support the conclusions?

Reviewer #1: Yes

Reviewer #2: Yes

Reviewer #3: Yes

Reviewer #4: Partly

2. Has the statistical analysis been performed appropriately and rigorously? 

Reviewer #1: N/A

Reviewer #2: Yes

Reviewer #3: No

Reviewer #4: Yes

3. Have the authors made all data underlying the findings in their manuscript fully available?

Reviewer #1: Yes

Reviewer #2: Yes

Reviewer #3: Yes

Reviewer #4: Yes

4. Is the manuscript presented in an intelligible fashion and written in standard English?

Reviewer #1: Yes

Reviewer #2: Yes

Reviewer #3: Yes

Reviewer #4: Yes

5. Review Comments to the Author

Reviewer #1: The study optimizes glabridin use in skin whitening applications by using CD-encapsulated glabridin, which reduces variation and improves water solubility, stability, and bioactivity. This provides insights for future studies and lays a foundation for safer, more effective whitening products. It is a good effort to find complex information of science required to correctly evaluate the cytotoxicity and effectiveness.

Reviewer #2: The manuscript is well written and contains some important information therefore i recommend its publication. However authors need to carefully revise the whole manuscript to avoid any grammatical and spells mistake.

Reviewer #3: The manuscript describes a comparative study that addresses a crucial concern related to safe-usage of glabridin, used as a whitening agent. Importantly, authors have evaluated cytotoxicity mediated by glabridin, if dissolved DMSO. Further, as an alternative to DMSO, authors showed that cyclodextrin (CD) mediated encapsulation, serve as better solvent with reduced cytotoxity and increased whitening efficiency.

However, it should also be taken into account that the importance of cyclodextrin in terms of solubility and bioavailability is already studied.

Importantly, authors MUST explain/rectify/address following major/minor comments for a comprehensive and conclusive study.

1. Statistical significance is not shown in most of the data. Kindly provide all the data with proper significance.

2. Line 168-169: Authors mentioned that “glabridin enter the hydrophobic cavity of CD, getting trapped within the CD molecule” Please mention the reference for the mechanism of CD mediated solubility of glabridin.

3. Line 169: Authors mentioned that “HPLC revealed that the encapsulation rate of glabridin in CD was 9.73%”. Please provide the reference for the statement.

4. Line 172: Authors mentioned that “maximum glabridin concentration typically used in cytotoxicity assays is less than 100 µg/mL”. It requires appropriate reference.

5. Figure 1: Need to provide the reference for the diagram showing CD-mediated encapsulation of glabridin.

6. Line 193: Need to correct the figure reference in the text from 2c to 2b.

7. Figure 3: The turbidity images and microscopy images for precipitation and crystals are blurred. Authors should provide high quality and clear images for the same.

8. Line 219-222: Authors speculated two consequences for decreased bioavailability. However, these speculations either should be backed up by some data or must have appropriate references. Same also goes for line 240-252, author should provide data or reference to support physical damage by crystals and their transport.

9. Figure 4a, 4b and 4c: Some of error bars shown in these figures does not coincides with graphs. It is unlikely that errors bars get separated from main graph, if generated using trusted scientific software/tools such as Graphpad Prism. Authors should provide images with better quality.

10. Figure 4: Cell viability assay should also include controls i.e. effect of DMSO and cyclodextrin alone. This is important to ensure that the observed cell viability is not contributed by the solvents (DMSO or CD) itself.

11. Please provide the reference for Melanin synthesis inhibition assay OR author should provide data to validate the usability of the assay. Also provide the protocol for measuring melanin synthesis “rate”.

12. Figure 5: Authors should provide detailed figure legend with the abbreviations (NC and BC etc.) used in the figures. It is difficult to interpret data without it, as it is also not mentioned anywhere in the text.

Reviewer #4: Strengths of the Paper

Novel Insight into Glabridin Solubility:

The study provides a fresh perspective on how the poor solubility of glabridin in DMSO impacts its cytotoxicity and whitening efficacy. This is a significant contribution to the field of cosmetic science, as it challenges existing methodologies and interpretations in cytotoxicity assays.

The use of cyclodextrin (CD) encapsulation as an alternative to enhance solubility and reduce cytotoxicity is innovative and practical.

Clear Experimental Focus:

The manuscript clearly defines its objectives, which are to reassess the cytotoxicity and efficacy of glabridin by addressing solubility issues. This focus ensures that the study remains relevant to both academic research and industry applications.

Relevance to Cosmetic Applications:

By linking the findings to the development of safer and more effective whitening products, the study has direct implications for the cosmetics industry. This applied aspect enhances its utility.

Comprehensive Background:

The introduction provides a detailed overview of glabridin's properties, its role in cosmetics, and the challenges associated with its use, setting a solid foundation for the study.

Weaknesses and Areas for Improvement

Limited Experimental Scope:

The study focuses primarily on solubility in DMSO and CD encapsulation but does not explore other potential solvents or delivery systems that could address similar issues. Broadening the scope to include alternative methods could strengthen the conclusions.

Insufficient Quantitative Data:

While the paper discusses the formation of crystals and their impact on cytotoxicity, it lacks detailed quantitative data on crystal size, shape, or concentration. Including such data would provide stronger evidence for the claims about physical damage caused by sharp crystals.

Lack of In Vivo Validation:

The study relies solely on in vitro assays to evaluate cytotoxicity and whitening efficacy. Incorporating in vivo experiments (e.g., animal models or human skin tests) would enhance the applicability of the findings to real-world conditions.

Potential Bias from Industry Funding:

The research is funded by Guangzhou Fanzhirong Cosmetics Co., Ltd., which may introduce a conflict of interest since the findings directly benefit the cosmetics industry. Although the authors declare no competing interests, independent replication of results would increase credibility.

Overgeneralization of Results:

The conclusion that previous studies have overestimated cytotoxicity and underestimated whitening efficacy may be too broad without directly comparing multiple prior studies under similar experimental conditions.

Clarity in Methodology:

The methodology section lacks sufficient detail about how CD encapsulation was performed and how its efficacy was measured compared to DMSO-based methods. Providing more technical details would improve reproducibility.

Ethics Statement Absence:

Although no human or animal subjects were involved, explicitly stating this in an ethics statement would align with best practices for transparency.

Suggestions for Improvement

Include more quantitative data on crystal formation (e.g., size distribution, morphology) using microscopy or other imaging techniques.

Explore additional solvents or formulations beyond CD encapsulation to provide a broader solution framework.

Conduct in vivo studies to validate findings under conditions that mimic real-world cosmetic applications.

Address potential biases by encouraging independent replication or collaboration with academic institutions.

Expand on methodological details, particularly regarding CD encapsulation protocols and comparative analysis techniques.

Provide a more balanced discussion by acknowledging limitations in generalizing findings across all prior studies.

Overall Assessment

The paper presents valuable insights into how solubility affects glabridin's properties, making it relevant for both researchers and practitioners in cosmetics science. However, its impact could be significantly enhanced by addressing methodological gaps, incorporating quantitative analyses, validating results through in vivo studies, and mitigating potential biases from industry funding sources.

6. PLOS authors have the option to publish the peer review history of their article (what does this mean? ). If published, this will include your full peer review and any attached files.

**Do you want your identity to be public for this peer review?** For information about this choice, including consent withdrawal, please see our Privacy Policy .

Reviewer #1: No

Reviewer #2: **Yes: ** Shakil Ahmad

Reviewer #3: No

Reviewer #4: No

---

## [Author Response · Author response to Decision Letter 0]

27 Apr 2025

Please see the detailed information of our response to reviewers and editor in the uploaded file named "Response to Reviewers"

---

## [Decision Letter · Decision Letter 1]

12 May 2025

Overestimated cytotoxicity and underestimated whitening efficacy of glabridin: A result of its poor solubility in DMSO

PONE-D-25-01303R1

Dear Dr. Sen Hou,

We’re pleased to inform you that your manuscript has been judged scientifically suitable for publication and will be formally accepted for publication once it meets all outstanding technical requirements.

Kind regards,

Yusuf Ahmed Haggag, PhD

Academic Editor

PLOS ONE

Additional Editor Comments (optional):

Reviewers' comments:

Reviewer's Responses to Questions

**Comments to the Author**

1. If the authors have adequately addressed your comments raised in a previous round of review and you feel that this manuscript is now acceptable for publication, you may indicate that here to bypass the “Comments to the Author” section, enter your conflict of interest statement in the “Confidential to Editor” section, and submit your "Accept" recommendation.

Reviewer #4: All comments have been addressed

2. Is the manuscript technically sound, and do the data support the conclusions?

Reviewer #4: Yes

3. Has the statistical analysis been performed appropriately and rigorously? 

Reviewer #4: Yes

4. Have the authors made all data underlying the findings in their manuscript fully available?

Reviewer #4: Yes

5. Is the manuscript presented in an intelligible fashion and written in standard English?

Reviewer #4: Yes

6. Review Comments to the Author

Reviewer #4: (No Response)

7. PLOS authors have the option to publish the peer review history of their article (what does this mean? ). If published, this will include your full peer review and any attached files.

**Do you want your identity to be public for this peer review?** For information about this choice, including consent withdrawal, please see our Privacy Policy .

Reviewer #4: No

---

## [Editor Report · Acceptance letter]

PONE-D-25-01303R1

PLOS ONE

Dear Dr. Hou,

I'm pleased to inform you that your manuscript has been deemed suitable for publication in PLOS ONE. Congratulations! Your manuscript is now being handed over to our production team.

Kind regards,

on behalf of

Dr. Yusuf Ahmed Haggag

Academic Editor

PLOS ONE